# Inhibitory Effects of Roseoside and Icariside E4 Isolated from a Natural Product Mixture (No-ap) on the Expression of Angiotensin II Receptor 1 and Oxidative Stress in Angiotensin II-Stimulated H9C2 Cells

**DOI:** 10.3390/molecules24030414

**Published:** 2019-01-23

**Authors:** Eun Young Hong, Tae Yang Kim, Gwan Ui Hong, Hanna Kang, Jung-Yun Lee, Jae Yeo Park, Se-Chan Kim, Young Ho Kim, Myung-Hee Chung, Young-In Kwon, Jai Youl Ro

**Affiliations:** 1Life & Science Research Center, Hyunsung Vital Co. Ltd., Seoul 07255, Korea; h_medical@naver.com (E.Y.H.); gwanuihong@gmail.com (G.U.H.); legolas018@naver.com (J.Y.P.); 2Department of Food and Nutrition, Hannam University, Daejeon 34054, Korea; xodid5606@naver.com (T.Y.K.); hanna9506@hanmail.net (H.K.); 3Natural Products Institute, Proteinworks Co. Ltd., Daejeon 07255, Korea; seembeeks@hanmail.net; 4Department of Bio Quality Control, Korea Bio Polytechnic, Chungnam 32943, Korea; sechage@kopo.ac.kr; 5Department of Pharmacy, Choongnam National University, Daejeon 34134, Korea; yhk@cnu.ac.kr; 6Gil Hospital and Lee Gil Ya Cancer & Diabetes Institute, Gachon University, Inchon 21999, Korea; mhchung@gachon.ac.kr; 7Department of Pharmacology, Sungkyunkwan University School of Medicine, Suwon 03063, Korea

**Keywords:** cardiomyocytes (H9C2 cells), angiotensin II, hypertension, No-ap (natural product mixture), reactive oxygen species, roseoside, icariside E4

## Abstract

Hypertension is a major risk factor for the development of cardiovascular diseases. This study aimed to elucidate whether the natural product mixture No-ap (NA) containing *Pine densiflora*, *Annona muricate*, and *Monordica charantia*, or its single components have inhibitory effects on hypertension-related molecules in Angiotensin II (Ang II)-stimulated H9C2 cells. Individual functional components were isolated and purified from NA using various columns and solvents, and then their structures were analyzed using ESI–MS, ^1^H-NMR, and ^13^H-NMR spectra. H9C2 cells were stimulated with 300 nM Ang II for 7 h. NA, telmisartan, ginsenoside, roseoside (Roseo), icariside E4 (IE4), or a combination of two components (Roseo and IE4) were administered to the cells 1 h before Ang II stimulation. The expression and activity of hypertension-related molecules or oxidative molecules were determined using RT-PCR, western blot, and ELISA. Ang II stimulation increased the expression of Ang II receptor 1 (AT1), tumor necrosis factor-α (TNF-α), monocyte chemoattractant protein-1 (MCP-1), tumor growth factor-β (TGF-β) mRNA, and nicotinamide adenine dinucleotide phosphate (NADPH) oxidase activity and the levels of hydrogen peroxide (H_2_O_2_) and superoxide anion (•O_2_^−^) and reduced anti-oxidant enzyme activity. NA significantly improved the expression or activities of all hypertension-related molecules altered in Ang II-stimulated cells. Roseo or IE4 pretreatment either decreased or increased the expression or activities of all hypertension-related molecules similar to NA, but to a lesser extent. The pretreatment with a combination of Roseo and IE4 (1:1) either decreased or increased the expression of all hypertension-related molecules, compared to each single component, revealing a synergistic action of the two compounds. Thus, the combination of single components could exert promising anti-hypertensive effects similar to NA, which should be examined in future animal and clinical studies.

## 1. Introduction

Hypertension is a major risk factor for the development of cardiovascular diseases including coronary artery disease, stroke, heart failure, peripheral vascular disease [1]. Angiotensin II (Ang II) is a vasoactive peptide of the renin–angiotensin system (RAS). The cellular effects of Ang II are mediated by at least two receptors, Ang II receptor 1 (AT1) and Ang II receptor 2 (AT2). Ang II, through AT1 or AT2, plays a key role in blood pressure homeostasis [2]. Ang II binds AT1 to induce NADPH oxidase activation and leads to the production of reactive oxygen species (ROS) [3,4,5]. Thus, the increasing of ROS is involved in cardiovascular diseases-related changes such as hypertrophy, fibrosis, tissue inflammation, or vascular remodeling in the heart [4,5,6].

Inhibition of the RAS through AT receptor blockers (ARBs) can prevent cardiovascular disease-related events [7]. ARBs (for example, losartan and telmisartan), which are a new class of approved anti-hypertensive agents, prevent the hypertensive effects of Ang II by the selective blockade of AT1 [8,9]. However, ARBs produce undesirable side effects such as headache, fatigue, and dizziness [7]. Thus, we have much interest in the search for natural products with anti-hypertensive effects and reduced side effects.

Our company (Hyunsung Vital Co. Ltd., Seoul, Korea) has manufactured a natural product complex (No-ap, NA) expected to downregulate blood pressure. NA contains three natural materials, i.e., *Pinus densiflora*, *Annona muricata* L., and *Monordica charantia*.

*P. densiflora* is widely distributed around the world, particularly in Korea and Japan [10]. Pine bark or needle have been reported to be effective scavengers of ROS [11,12] and to have a suppressive effect on the expression of pro-inflammatory mediators [13]. Pine needles have anti-hypertensive effects [14]. *A.a muricata* L., which is known as graviola or guanabana, is widely found in India, South and Central America, tropical West Africa, and Asia [15]. It has been reported that a decoction made from *A. muricata* can be used for hypertension therapy and that the plant extract has anti-oxidative and anti-hypertensive properties [16]. *M. charantia* L. is a common vegetable in Okinawa, where it has been recently used in the therapy of hypertension, diabetes, and dyslipidemia [17], and its phenolic extract has inhibitory properties against angiotensin-1-converting enzyme, hypertension, and oxidative stress [18]. Thus, these findings imply that NA may have anti-hypertensive effects, as it is also supported by its wide use as a folk remedy and by laboratory experiments [14,15,16]. 

Although there are reports that the different constituents of NA, i.e., *P. densiflora* needle, *A. muricata*, and *M. charantia* may have anti-hypertensive properties, the single components of this mixture have not been isolated and examined [14,15,16]. This study investigated for the first time whether NA has inhibitory effects on the hypertension-related molecules in Ang II-stimulated H9C2 cells. After confirmation of the anti-hypertensive effects, this study aimed to identify the single functional components of NA and to investigate whether they have anti-hypertensive properties individually. We observed that the pretreatment with a combination of roseoside and icariside E4, which showed strong activity among the five single components identified in NA, had anti-hypertensive effects by downregulating ROS generated via the expression of AT1 and the activity of NADPH oxidase.

## 2. Results

### 2.1. Effects of NA on the Expression of Hypertension-Related Molecules in Ang II-Stimulated H9C2 Cells

AT1 is an important effector controlling blood pressure (BP) and blood volume in the cardiovascular system [3]. We first examined the effects of NA on AT1 expression in Ang II-stimulated H9C2 cells. AT1 expression was increased in the Ang II-stimulated H9C2 cells, compared with negative control (NC, treated with phosphate-buffered saline) cells. NA (60, 100, 200 μg/mL) reduced AT1 expression in a dose-dependent manner (Figure 1A). A high dose of NA reduced AT1 expression similar to telmisartan (Telmis), which is known as an AT1 blocker preventing Ang II-induced oxidative stress and vascular remodeling in hypertension [9]. Pretreatment with 200 μg/mL (corresponding to the high dose of NA) of ginsenoside (Gin), which was used as one of the natural positive controls for the natural product mixture (NA), had no effect on AT1 expression in Ang II-stimulated H9C2 cells. 

It has been reported that inflammation has a crucial role in the pathogenesis of hypertension [19,20,21]. The inflammatory process, with ROS generation and increase in cytokines’ releases, is a hallmark of hypertension [20,21]. Thus, in order to investigate whether NA prevents inflammation in Ang II-stimulated H9C2 cells, the expression of inflammatory cytokines was examined. The expression levels of tumor necrosis factor-α (TNF-α), monocyte chemoattractant protein-1 (MCP-1), and tumor growth factor-β (TGF-β) were increased in Ang II-stimulated H9C2 cells (Figure 1A). NA pretreatment significantly suppressed the expression of these inflammatory cytokines caused by Ang II in a dose-dependent manner. A high dose of NA showed stronger inhibitory responses than Telmis. Gin did not show any effects in all cases. Therefore, hereafter, we will not consider Gin effects.

### 2.2. Effects of NA on Oxidative Stress in Ang II-Stimulated H9C2 Cells

It has been reported that Ang II-induced hypertension by ROS generated via nicotinamide adenine dinucleotide phosphate (NADPH) oxidase [22,23]. Thus, we examined the effects of NA on NADPH oxidase activity in Ang II-stimulated H9C2 cells. NA pretreatment reduced NADPH oxidase activity in Ang II-stimulated H9C2 cells (Figure 1B). 

Next, we examined the effects of NA on the generation of ROS or on anti-oxidant enzyme activity in Ang II-stimulated H9C2 cells. The production of hydrogen peroxide (H_2_O_2_) or superoxide anion (•O_2_^−^) increased by Ang II stimulation was diminished in NA-pretreated cells in a dose-dependent manner (Figure 1C,D). The activities of catalase or superoxide dismutase (SOD) were decreased in Ang II-stimulated H9C2 cells. NA pretreatment increased anti-oxidant enzyme activity (catalase and SOD) in Ang II-stimulated H9C2 cells in a dose-dependent manner, similar to Telmis (Figure 1E).

### 2.3. Purification and Identification of Bioactive Ingredients in NA

As a natural product mixture, NA contains *P. densiflora* (75.0%), *A. muricata* (12.5%), *M. charantia* (12.5%). NA was extracted with methanol and successively partitioned with ethyl acetate and n-BuOH to afford ethyl acetate, n-BuOH, and water fractions. The n-BuOH fraction was subjected to silica gel and YMC RP-18 silica gel column chromatography, and five compounds (**1**–**5**) were identified. Their spectroscopic data and comparisons with previous data confirmed that these compounds were roseoside (**1**) [24], isolariciresinol 9-*O*-β-d-xyloside (**2**) [25], massonianoside B (**3**) [26], icariside E4 (**4**) [27], and nicotiflorin (**5**) [28] (Figure 2, Appendix A).

### 2.4. Effects of the Isolated Components Roseoside and Icariside E4 on the Expression of Hypertension-Related Molecules in Ang II-Stimulated H9C2 Cells

We first confirmed that two single components [roseoside (Roseo) and icariside E4 (IE4)], among the five isolated components, had biological activity. Three components (isolariciresinol 9-*O*-β-d-xyloside, massonianoside B, and nicotiflorin) did not have any effects on the expression of AT1 (data not shown). Roseo or IE4 (20, 30, 50 μg/mL) pretreatment significantly reduced the expression of all hypertension-related molecules (mRNA and protein) in Ang II-stimulated H9C2 cells in a dose-dependent manner, compared to Ang II stimulation alone (Figure 3). Treatment with 70 μg/mL of Roseo or 100 μg/mL of IE4 reduced the expression of all hypertension-related molecules by approximately 48% and 50%, respectively, compared to Ang II stimulation. However, only Roseo showed downregulating activity at a dose above 70 μg/mL (data not shown). Both components showed inhibitory effects on the expression of all hypertension-related molecules, although neither one exhibited inhibitory effects similar to those of Telmis or NA. Thus, we tried a combination pretreatment. A ratio of 1:1 for the combination pretreatment of Roseo and IE4 was used in preliminary experiments. The combination of Roseo and IE4 in a 1:1 ratio (each used in a dose of 25 μg/mL; total dose, 50 μg/mL) reduced the expression of all hypertension-related molecules by approximately 50–72% in Ang II-stimulated H9C2 cells. Thus, the combination pretreatment of Roseo and IE4 showed a strong synergistic inhibitory effect on the expression of all hypertension-related molecules in Ang II-stimulated H9C2 cells. 

### 2.5. Respective Effects of Roseoside and Icariside E4 on Oxidative stress in Ang II-Stimulated H9C2 Cells

We examined the effects of Roseo or IE4 on NADPH oxidase activity in Ang II-stimulated H9C2 cells. Roseo or IE4 pretreatment reduced NADPH oxidase activity in Ang II-stimulated H9C2 cells (Figure 4A). 

Next, we examined the effects of Roseo and IE4 on the generation of ROS and on anti-oxidant enzyme activity in Ang II-stimulated H9C2 cells. The generation of H_2_O_2_ or •O_2_^−^ increased by Ang II stimulation was diminished after Roseo or IE4 pretreatment in a dose-dependent manner (Figure 4B,C). Both Roseo and IE4 pretreatments increased the anti-oxidant activity (catalase and SOD), which was decreased by Ang II-stimulated H9C2 cells (Figure 4D). Roseo pretreatment showed reduction of ROS and increase of anti-oxidant enzyme activity similar to Telmis, whereas IE4 pretreatment showed smaller effects than Roseo or Telmis. 

The combination pretreatment of Roseo and IE4 at a ratio of 1:1 (each used at the dose of 25 μg/mL; total dose, 50 μg/mL) reduced the activity of NADPH oxidase and the generation of H_2_O_2_ and •O_2_^−^ in a dose-dependent manner in Ang II-stimulated H9C2 cells. The combination pretreatment increased the activity of catalase or SOD in a dose-dependent manner. Thus, the combination pretreatment of Roseo and IE4 at a 1:1 ratio showed stronger synergistic inhibitory effects on the suppression of oxidative stress and on the increase of anti-oxidant enzyme activities in Ang II-stimulated H9C2 cells, compared to the single components (Figure 4).

## 3. Discussion

We demonstrated that the natural product complex NA suppresses the expression of AT1, the generation of ROS (H_2_O_2_ and •O_2_^−^), and the expression of inflammatory cytokines (TNF-α, MCP-1, and TGF-β) produced via NADPH oxidase, which are related to hypertension, and that it increases the expression of anti-oxidant enzymes (catalase and SOD) in Ang II-stimulated H9C2 cells. Our data also demonstrate that two components, Roseo and IE4, among five identified components isolated and purified from NA, whose structures were also analyzed in this study, have functional activities in hypertension.

The component Roseo isolated from various plants, including *A. muricata* contained in NA, has a variety of functional activities. It relaxes precontracted aortic rings in an endothelium-dependent manner [29], increases insulin secretion [30], inhibits rat liver microsomal glucose-β-phosphate [31], potentiates the inhibitory activity against angiotensin-converting enzyme, although itself shows no activity against an its enzyme [32], shows inhibitory effects on lipopolysaccharide (LPS)-induced nitric oxide (NO) production in RAW264.7 cells [33], prevents oxidative stress [34], and has a depigmentation effect in melanocytes by inhibiting melanin synthesis [35]. However, it has not been reported yet that Roseo directly suppresses hypertensive effects caused by the upregulation of ROS.

The other component, IE4, isolated and purified from plants, such as *Ulmus pumila* L. and *Tabebuia roseo-alba*, has various functional activities. It inhibits NO production [36], shows anti-oxidant activity [37], demonstrates anti-nociceptive activity in a chemical pain-induced model [38], and may be beneficial in the traditional treatment of Alzheimer’s disease by preventing blood–brain barrier damage and inflammatory cell infiltration into the brain [39]. IE4, purified from pine trees, such as *P. densiflora*, *Pinus thunbergii*, and *Pinus morrisonicola* Hayata, also delays the coagulation time by inhibiting thrombin activity [40] and inhibits the release of β-hexosaminidase in RBL-2H3 cells [41]. However, it has not been reported yet that IE4 purified from *P. densiflora*, which is the most abundant component of NA, directly suppresses the hypertensive effects caused by the upregulation of ROS. There is only a report indicating that IE4 isolated from *P. morrisonicola* H. could be a promising anti-hypertensive candidate by blocking voltage-operating Ca^2+^ channels [42]. 

Hypertension is the most common cardiovascular risk factor. Ang II, which is known as one of many factors causing cardiovascular injury in hypertension, elicits many pathophysiological actions by inducing ROS generation via the activation of vascular NADPH oxidase [20,22,23,43]. Ang II stimulation can increase blood pressure in association with immune response activation and inflammation [21]. Infiltration of inflammatory cells into areas around blood vessels that occurs simultaneously with other events of the inflammatory process, such as the increase of ROS generation and of the levels of cytokines and chemokines, is a hallmark of hypertension [9,21,22]. Thus, our data suggest that NA may suppress blood pressure and vascular remodeling in hypertension mainly caused by RAS through downregulating of ROS produced via AT1 expression and NADPH oxidase activity, as demonstrated by the data showing that NA reduced hypertensive responses through inhibiting downstream pathways via AT1 expression and NADPH oxidase (which generates ROS), and then the generated ROS increased the release of the cytokine TNF-α and the chemokines MCP-1, which is known to induce inflammation and cell infiltration [44], respectively, and increased TGF-β, which is known to induce vascular remodeling in hypertension [9,22]. In addition, our data indicate that NA increased endogenous anti-oxidant enzymes (catalase and SOD) in Ang II-stimulated H9C2 cells. The endogenous anti-oxidant glutathione peroxidase (GPx), which scavenges H_2_O_2_, is not affected by Ang II stimulation in cardiac fibroblasts [45] and vascular adventitial fibroblasts [46]. Thus, we did not check GPx activity because the catalase also scavenges H_2_O_2_. However, it is necessary to investigate the differences between cell types.

Our observations also suggest that the components Roseo and IE4, isolated and purified from NA, may independently suppress hypertension-related molecules such as ROS, cytokines and chemokines, and TGF-β, which are typically associated with oxidative stress, inflammation, and vascular remodeling in hypertension, by increasing endogenous anti-oxidant enzymes in Ang II-stimulated H9C2 cells. In addition, we also show that combinations of Roseo and IE4 (1:1 ratio) may have strong synergistic or additive effects. 

In conclusion, these in vitro results support the conclusion that the functional compounds Roseo and IE4 have significant anti-hypertensive effects on Ang II-stimulated H9C2 cells. Our data suggest that Roseo and IE4 from NA, which have anti-oxidant, anti-inflammatory, and anti-vascular remodeling properties in hypertension and less side effects than the whole mixure NA, have a potential as health functional food supplements for hypertension and should be further evaluated in animal and clinical models.

## 4. Materials and Methods

### 4.1. Materials

NA, a natural product mixture, was donated by Hyunsung Vital Co. Ltd. (Seoul, Korea) which makes a variety of healthy functional foods. NA contains *P. densiflora* (75.0%), *A. muricata* (12.5%), and *M. charantia* (12.5%). These plants were extracted using hot water (90 °C) for 24 h, and the extract was evaporated by a Liquefied extractor (Hyunsung Vital Co. Ltd., Seoul, Korea) to yield a powder of NA. This powder was named NA (No-ap), natural product mixture.

### 4.2. Purification and Identification of Bioactive Ingredients

NA (1.9 kg) was extracted using methanol (4 L, 95%) at room temperature for 2 days. The methanol extract (556.4 g) was concentrated under pressure, dissolved in distilled water (2 L), and successively partitioned with ethyl acetate and n-BuOH to afford ethyl acetate (23.0 g, A), n-BuOH (122.1 g, B), and water fractions.

The n-BuOH extract (120 g) was separated by vacuum liquid chromatography using a silica gel column with a gradient solvent mixture of CHCl_3_–MeOH (30:1, 25:1, 20:1, 15:1, 10:1, 8:1, 6:1, 4:1, 2:1, 1:1, and 100% MeOH) to afford 11 subfractions (B-1 to B-11). Next, subfraction B-6 (2.1 g) was subjected to YMC RP-C18 silica gel column and was eluted with a solvent mixture of MeOH–H_2_O (1:2), yielding compound 1 (20.0 mg), compound 2 (10.0 mg), compound 3 (30.0 mg), and compound 4 (10.0 mg). Further purification of subfraction B-9 via YMC RP-C18 silica gel column, using mixtures of MeOH−H_2_O (1:1.5), and preparative HPLC yielded compound 5 (10.0 mg). 

### 4.3. Cardiomyocytic H9C2 Cell Line Culture

The rat cardiomyocytic H9C2 cell line (H9C2 cells) was obtained from the Korean Cell line Bank (KCLB, Seoul, Korea). H9C2 cells were grown in Dulbecco’s modified eagle medium (DMEM) (HyClone, Logan, UT, USA) supplemented with 1% l-glutamine, 1% antibiotic penicillin/streptomycin solution (Sigma-Aldrich, St. Louis, MO, USA), and 10% fetal bovine serum (HyClone, Logan, UT, USA). H9C2 cells were maintained at 37 °C in a humidified atmosphere with CO_2_, and the media were replaced every 3 days [47].

### 4.4. Cell line Stimulation and Treatment

H9C2 cells (1 × 10^6^ cells) were stimulated with 300 nM angiotensin II (Ang II; Sigma-Aldrich, St. Louis, USA) and then incubated for 7 h [47]. The cells were centrifuged (470× *g*, 3 min) to separate the supernatants and pellet the cells. The cells were used to determine the expression of all molecules related to hypertension. NA (60, 100 or 200 μg/mL), telmisartan (Telmis; 10 μM), ginsenoside (Gin; 200 μg/mL), roseoside (Roseo; 20, 30 or 50 μg/mL), icariside E4 (IE4; 20, 30 or 50 μg/mL), the last two as purified components isolated from NA, and combinations of Roseo and IE4 (1:1 ratio, 20, 30, or 50 μg/mL) were administered to the cells 1 h before Ang II stimulation. As a negative control (NC), phosphate-buffered saline (PBS) was used, which was also Ang II solvent. Telmis, which is a drug used in the clinic, was used as a positive control. The optimal concentrations of Ang II stimulation, NA, Gin, Roseo, and IE4 were determined in preliminary experiments (data not shown). Gin was used as a natural positive control for the natural product mixture (NA).

### 4.5. Reverse Transcription-Polymerase Chain Reaction (RT-PCR)

Total mRNA was isolated from H9C2 cells (1 × 10^6^ cells) using TRIzol reagent (Invitrogen, Life Technologies Ltd., Waltham, CA, USA). RT-PCR was performed in a final volume of 20 μL, using a high-capacity cDNA Reverse Transcription kit (Applied Biosystems, Foster City, CA, USA) and a G-taq kit (Cosmogenetech, Seoul, Korea) in an automated thermal cycler (Bio-Rad, Laboratories, Hercules, CA, USA). The PCR assays were performed for 35 cycles. Each cycle consisted of the following steps: denaturation at 94 °C for 30 s, annealing at 51 °C for 45 s, and extension at 72 °C for 1 min. The results were expressed as a ratio to GAPDH mRNA. The PCR products were analyzed using 1% agarose gel and visualized under UV light after staining with StaySafe nucleic acid gel stain (Real Biotech Corporation, Banqiao, Taiwan) [48].

The primer sequences used were as follows: AT1 sense, 5′-CAT AGG ACT GGG CCT AAC CA-3′; AT1 anti-sense, 5′-GCC GTA AAA CAG AGG GTT CA-3′; TNF-α sense, 5′-TTC TGT CCC TTT CAC TCA CTG G-3′; TNF-α anti-sense, 5′-TTG GTG GTT TGC TAC GAC GTG G-3′; MCP-1 sense, 5′-GAA GGA ATG GGT CCA GAC AT-3′; MCP-1 anti-sense, 5′-ACG GGT CAA CTT CAC ATT CA-3′; TGF-β sense, 5′-CTC TCC ACC TGC AAG ACC AT-3′; TFG-β anti-sense, 5′-CTG CCG TAC AAT TCC AGT GA-3′; GAPDH sense, 5′-AAC TTT GGC ATT GTG GAA GG-3′; GAPDH anti-sense, 5′-ACA CAT TGG GGG TAG GAA CA-3′.

### 4.6. Determination of the Activity of NADPH Oxidase, Superoxide Dismutase, and Catalase and the Levels of Hydrogen Peroxide and Superoxide Anion

The activities of NADPH oxidase, catalase, and SOD, and the amounts of H_2_O_2_ and •O_2_^−^ were measured in the lysates of H9C2 cells stimulated with Ang II using a NADPH oxidase assay kit, a H_2_O_2_ assay kit (Abcam, Cambridge, UK), a •O_2_^−^ assay kit, a SOD assay kit (Cell Biolabs. Inc., San Diego, CA, USA), and a catalase activity kit (Biovision, Milpitas, CA, USA), respectively. Briefly, H9C2 cells (1 × 10^6^ cells) were washed three times with PBS and placed in lysis buffer (PBS containing 1% Triton X-100). Standard diluents (100 μL), lysates obtained from each sample (100 μL), and 100 μL of reaction mixture (50 μL enzyme working solution and 50 μL probe) were added in 96-well plates; the plates were then incubated on a plate shaker at room temperature for the time periods indicated (30 min for NADPH oxidase, SOD, and catalase; 1 h for H_2_O_2_ and •O_2_^−^). The optical density was read at 450 nm for NADPH oxidase, •O_2_^−^, and SOD, and at 590 nm for H_2_O_2_ and catalase. Standard curves were made using serial dilutions of a standard sample, and then the activity was calculated according to the manufacturer’s instructions. The lowest detection limit for NADPH oxidase was below 20 pg/mL, for H_2_O_2_ it was 0.04 pmol/μL, for •O_2_^−^ and SOD it was 1.2 mU/μL, and for catalase it was 0.078 pg/mL.

### 4.7. Western Blot Analysis 

H9C2 cells (1 × 10^6^ cells) harvested from Ang II-stimulated cells were suspended in a low-salt lysis buffer [50 mM Tri-HCl (pH 7.9), 1.0 mM EDTA, 150 mM NaCl, 1.0% NP40, 5 mM NaF, 0.25% Na deoxycholate, 2 mM NaVO_3_, protease inhibitors cocktail] and allowed to swell on ice for 30 min. The cells were then homogenized using a micropipette. After centrifugation, the supernatants obtained from the cell extracts were analyzed by 10% SDS-polyacrylamide gel electrophoresis and electrophoretically transferred to nitrocellulose membranes (Amersham Biosciences, Piscataway, NJ, USA). The membranes were washed with PBS containing 0.1% Tween 20 (PBST) and then blocked for 1 h in PBST containing 5% skim milk. After washing the membranes with PBST, they were treated with primary antibodies against actin, AT1, TNF-α (Cell Signaling Technology, Beverly, MA, USA), MCP-1, or TGF-β (Abcam, Cambridge, UK) diluted with PBST (1:1000). The membranes were washed with PBST and treated with horseradish peroxidase (HRP)-conjugated goat anti-mouse or HRP-conjugated goat anti-rabbit IgG (diluted to 1:5000) (Bethyl Laboratories, Montgomery, TX, USA) in PBST for 1 h. After washing, the protein bands were visualized by Enhanced Chemi-Luminescence (ECL; Amersham Biosciences, Piscataway, NJ, USA) using a chemiluminometer (Bio-Rad, Laboratories, CA, USA) [48].

### 4.8. Statistical Analysis

The experimental data are shown as means ± SEM (*n* = 4). The unpaired Student’s *t*-test was used to compare two groups. Multiple-group comparisons were performed using two-way ANOVA followed by Scheffe’s post-hoc test, using the SPSS software (SPSS Inc., Chicago, IL, USA). Values of *p* < 0.05 were considered to indicate statistical significance. Densitometry analyses of Western blots and RT-PCR were performed with Quantity One (version 4.6.3; Bio-Rad, Hercules, CA, USA) and the results are indicated as means ± SEM (*n* = 4), obtained from the ratio of each band density to those of the control and the loading control of four independent experiments.

## Figures and Tables

**Figure 1 molecules-24-00414-f001:**
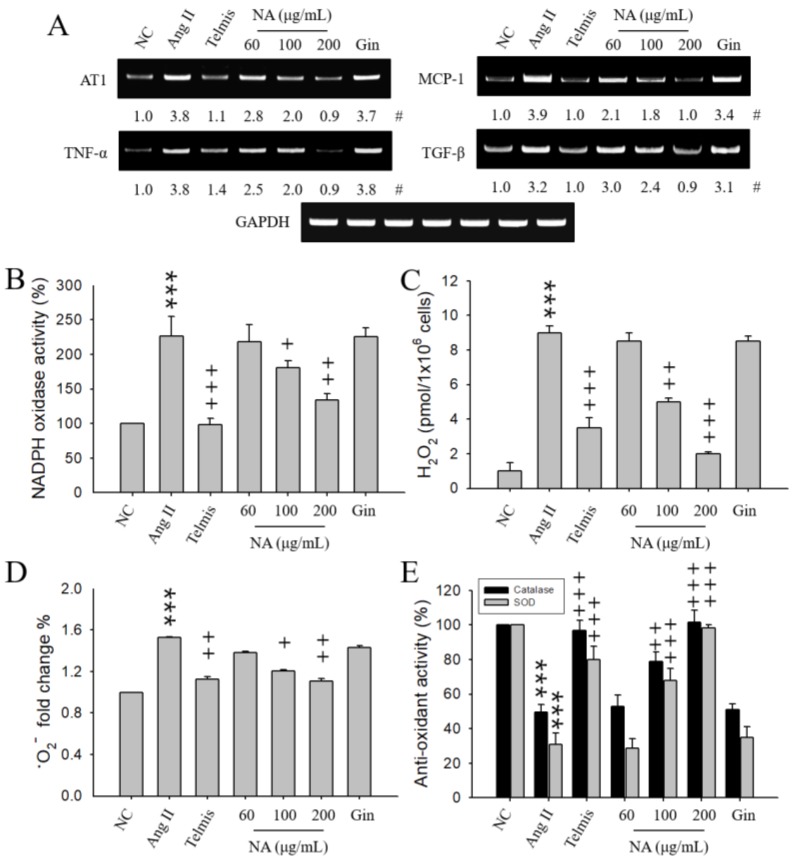
Effects of the natural product mixture (No-ap, NA) on the expression of hypertension-related molecules or oxidative stress in the Ang II-stimulated H9C2 cells. H9C2 cells (1 × 10^6^ cells) were stimulated with 300 nM Ang II for 7 h. No-ap (NA), telmisartan (Telmis), or ginsenoside (Gin) were administered 1 h before Ang II stimulation. The expression of AT1, TNF-α, MCP-1, TGF-β was determined in mRNA extracts isolated from H9C2 cells using RT-PCR. The activity of NADPH oxidase, catalase, and SOD, and the amounts of H_2_O_2_ and •O_2_^−^ were determined in cell lysates isolated from H9C2 cells using an ELISA kit. The reactions were analyzed using an ELISA plate reader at 450 nm for the activities of NADPH oxidase and SOD and •O_2_^−^ amounts, and at 590 nm for H_2_O_2_ amounts and catalase activity. (**A**) Expression of AT1 and cytokines. (**B**) Activity of NADPH oxidase. (**C**) Amounts of H_2_O_2_. (**D**) Amounts (fold change %) of •O_2_^−^. (**E**) Activities of catalase and SOD. #, Numbers below the band images, indicating the mean values (*n* = 4 independent experiments) obtained from the ratio of the band density of each group to those of the corresponding controls and loading control GAPDH. The results represent the mean ± SEM (*n* = 4) obtained from four independent experiments performed in triplicates. NC, negative control; Ang II, angiotensin II stimulation; AT1, angiotensin II receptor 1; TNF-α, tumor necrosis factor-α; MCP-1, monocyte chemoattractant protein-1; TGF-β, tumor growth factor-β; NADPH, nicotinamide adenine dinucleotide phosphate; H_2_O_2_, hydrogen peroxide; •O_2_^−^, superoxide anion; SOD, superoxide dismutase. ***, *p* < 0.001 versus the NC. ^+^, *p* < 0.05; ^++^, *p* < 0.01; ^+++^, *p* < 0.001 versus the Ang II stimulation.

**Figure 2 molecules-24-00414-f002:**
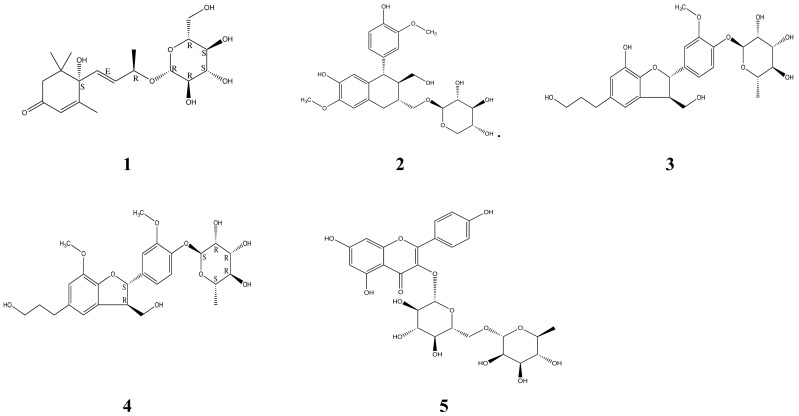
Chemical structures of the compounds isolated from NA. NA (1.9 kg) was fractioned using various organic chemicals and columns as described in “Materials and Methods”. Finally, five single components were identified from NA, and then the structure of each component was analyzed using ESI–MS, ^1^H-NMR, and ^13^C-NMR spectra. The inhibitory effects on the expression of AT1 of each fraction separated with organic chemicals were determined (data not shown). Structures of (**1**), (**2**), (**3**), (**4**), and (**5**) indicate roseoside, isolariciresinol 9-*O*-β-d-xyloside, massonianoside B, icariside E4, and nicotiflorin, respectively. The molecular formulas of the components roseoside and icariside E4, which demonstrated biological activity, re C19H30O8 (MW, 386.1941) and C26H34O10 (MW, 506.5480), respectively.

**Figure 3 molecules-24-00414-f003:**
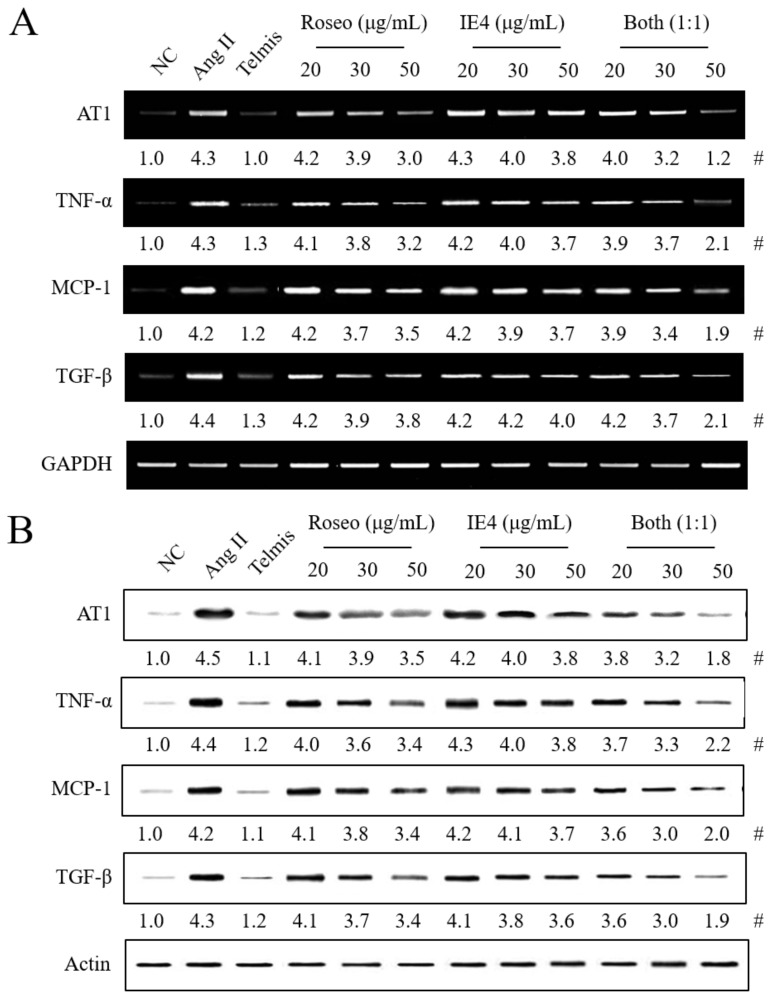
Effects of roseoside and icariside E4, alone or in combination, on the expression of hypertension-related molecules in the Ang II-stimulated H9C2 cells. The experimental details of the stimulation of H9C2 cells and their treatment with different compounds were described in Figure 1 legend. Telmis (10 μM), roseoside (Roseo; 20, 30, or 50 μg/mL), icariside E4 (IE4; 20, 30, or 50 μg/mL), and Roseo/E4 combinations (20, 30, or 50 μg/mL, 1:1 ratio) were administered 1 h before Ang II stimulation. The expression of all hypertension-related molecules was determined in mRNA or protein extracts isolated from H9C2 cells using RT-PCR and Western blot, respectively. (**A**) Expression of all hypertension-related molecules’ mRNA. (**B**) Protein expression of all hypertension-related molecules. Both, combination of Roseo and IE4 (1:1 ratio; each component used at a dose of 10, 15, 25 μg/mL; total doses for the different mixtures at a 1:1 ratio were 20, 30, 50 μg/mL). Ang II, angiotensin II stimulation. #, Numbers below the band images indicate the mean values (*n* = 4) obtained as described in Figure 1 legend.

**Figure 4 molecules-24-00414-f004:**
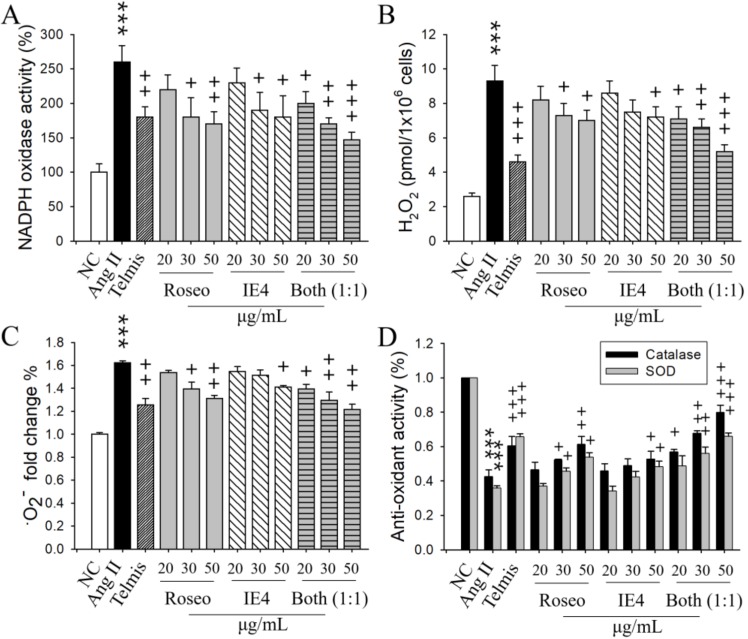
Effects of roseoside, icariside E4, and their combination on oxidative stress in Ang II-stimulated H9C2 cells. The experimental details of the stimulation of H9C2 cells and their treatment with different compounds were described in Figure 1 legend. Telmis (10 μM), Roseo (20, 30, or 50 μg/mL), IE4 (20, 30, or 50 μg/mL), and combinations of Roseo and IE4 (20, 30, or 50 μg/mL, 1:1 ratio) were administered to the cells 1 h before Ang II stimulation. The activities of NADPH oxidase, catalase, and SOD, and the amounts of H_2_O_2_ and •O_2_^−^ were determined in cell lysates isolated from H9C2 cells using an ELISA kit. The reactions were analyzed using an ELISA plate reader as described in Figure 1 legend. (**A**) Activity of NADPH oxidase. (**B**) Amounts of H_2_O_2_. (**C**) Amounts (fold change %) of •O_2_^−^. (**D**) Activities of catalase and SOD. The results represent the mean ± SEM (*n* = 4) obtained from four independent experiments performed in triplicates. ***, *p* < 0.001 versus NC. ^+^, *p* < 0.05; ^++^, *p* < 0.01; ^+++^, *p* < 0.001 versus Ang II stimulation.

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
