# Peer review of "Inhibitory Effects of Roseoside and Icariside E4 Isolated from a Natural Product Mixture (No-ap) on the Expression of Angiotensin II Receptor 1 and Oxidative Stress in Angiotensin II-Stimulated H9C2 Cells"

_molecules, 2019, doi:10.3390/molecules24030414_

Round 1

Reviewer 1 Report

In the present work, Authors Eun Young Hong et al investigated the impact of specific compounds  (roseoside   and   icariside   E4)   isolated in natural product mixture on the AT1  receptor expression and oxidative stress in angiotensin  II-stimulated H9C2 cells.

As far as identification of bioactive molecules is concerned, this work deals about a subject of great interest. It has been performed by a capable group by using valuable techniques.

That said I have few points that should be addressed by the Authors:

-          English style should be improved: last sentence of the abstract is not correct and should be rephrased. Abstract line 31 no capital for western blot. “… oxidative  molecules  was  determined” : were determined.

-          Abstract line 36. Authors wrote “Roseo  or  IE4  pretreatment decreased/increased…” similarly two line after they wrote “Combination pretreatment (1:1) of the single component synergistically decreased/increased…”. To my point of view, Authors can not be so vague when summarizing their results. They should focus of a relevant effect of their preparation, if any (either decrease or increase).

-          English style should be improved. As an example, at the end of the introduction, Authors summarize their study “We observed  that NA contained two kinds (roseoside and icariside E4) among five single components had anti-hypertensive   effects  through  down-regulating  the  ROS   generation  caused  via  reducing  the  expression of AT1 and activity of NADPH oxidase.” Authors should include more punctuations and correct English style to render their message more clear cut.

-          It is not easy to understand what NA refers to. There are several definitions in the MS for NA:

It is specified “NA contains three natural materials Pinus densiflora, Annona muricata L. and Monordica charantia”. In the Method section it is specified hot water extraction were used to yield a powder of NA. Elsewhere is it specified as No-ap, natural product mixture.

-          Page 2 line 87. What is the cell treatment for the negative control (NC)?

-          Fig 1. Authors reported SOD and catalase antioxidant activities. It is important to include gluthathion peroxidase activity to have a complete overview of antioxidative enzyme activities.

-          Page 5 line 150. Authors wrote “We first confirmed that two single components [roseoside (Roseo) and icariside E4 (IE4)] among  five  components  had  activity  (data  not  shown).”

Did the Authors test the activity of the three other components?

Author Response

Manuscript ID: molecules-421944

Title: Inhibitory effects of reseoside and icariside E4 on the expression of angiotensin II receptor 1 and production of reactive oxygen species in angiotensin II-stimulated H9C2

Reviewer 1.

Comments and Suggestions for Authors

In the present work, Authors Eun Young Hong et al investigated the impact of specific compounds (roseoside and icariside E4) isolated in natural product mixture on the AT1 receptor expression and oxidative stress in angiotensin II-stimulated H9C2 cells.

As far as identification of bioactive molecules is concerned, this work deals about a subject of great interest. It has been performed by a capable group by using valuable techniques.

That said I have few points that should be addressed by the Authors:

1.  English style should be improved: last sentence of the abstract is not correct and should be rephrased. Abstract line 31 no capital for western blot. “… oxidative molecules was  determined”: were determined.

Answer: The last sentence of the "Abstract" is rephrased as "Thus, combination of single component could be promising anti-hypertensive effects of NA, which will be examined in animal and clinical studies" on page 1 line 43.

We changed to "were determined" and "western blot" on page 1 line 32.

2.  Abstract line 36. Authors wrote “Roseo or IE4 pretreatment decreased/increased…” similarly two line after they wrote “Combination pretreatment (1:1) of the single component synergistically decreased/increased…”. To my point of view, Authors can not be so vague when summarizing their results. They should focus of a relevant effect of their preparation, if any (either decrease or increase).

Answer: We changed them to "either decreased or increased" on page 1 line 38 and 41.

3.  English style should be improved. As an example, at the end of the introduction, Authors summarize their study “We observed  that NA contained two kinds (roseoside and icariside E4) among five single components had anti-hypertensive   effects  through  down-regulating  the  ROS generation caused via reducing  the  expression of AT1 and activity of NADPH oxidase.” Authors should include more punctuations and correct English style to render their message more clearly cut.

Answer: This sentence was revised to "We observed that combination pretreatment of roseoside and icariside E4, which showed strong functional activity among the five single components identified from NA, had anti-hypertensive effects through down-regulating the ROS generated via the expression of AT1 and activity of NADPH oxidase." on page 2 line 82-85.

4.  It is not easy to understand what NA refers to. There are several definitions in the MS for NA:

It is specified “NA contains three natural materials Pinus densiflora, Annona muricata L. and Monordica charantia”. In the Method section it is specified hot water extraction were used to yield a powder of NA. Elsewhere is it specified as No-ap, natural product mixture.

Answer: The definition for NA was revised as "NA contained Pinus densiflora (75.0%), Annona muricata (12.5%) and Monordica charantia (12.5%). These plants were extracted using hot water (90 °C) for 24 h, and evaporated by Liquefied extractor (Hyunsung Vital Co. Ltd, Seoul, Korea) to yield a powder of NA. This powder was named (or specified) as NA (No-ap), natural product mixture" in "Materials" on page 8 line 292-295.

5.  Page 2 line 91. What is the cell treatment for the negative control (NC)?

Answer: The phosphate buffered saline (PBS), which was used as angiotensin II solvent, was used as negative control (NC). It was described in "Results" on page 2 line 91 and "Materials" on page 9 line 322 and 323.

6.  Fig 1. Authors reported SOD and catalase antioxidant activities. It is important to include glutathione peroxidase activity to have a complete overview of antioxidative enzyme activities.

Answer: We agree with you. However, it takes too much times (approximately 5 weeks in Korea) to get commercialized kit for glutathione peroxidase activity. Thus, we could not get these results in time indicated from you.

Superoxide dismutase (SOD), catalase and glutathione peroxidase (GPx), which have developed in the number of tissues and cell types, are important endogenous antioxidant enzymes that scavenge intracellular ROS in physiological state. Overexpression of catalase or treatment with liposome-capsulated SOD prevents hypertension via reducing ROS generation [1]. Superoxide anion (×O2-) is produced by various enzymatic sources including NADPH oxidase, and dismutated spontaneously or enzymatically to hydrogen peroxide (H2O2) by SOD. The H2O2 can also be produced directly by NADPH oxidase. The H2O2 is subsequently and enzymatically reduced by glutathione peroxidase as well as by catalase (see Figure A) [2].

There are reports that Ang II decreased the mRNA and protein expression of SOD, but not those of catalase and GPx in cardiac fibroblasts of Wistar rats [3]. Ang II also decreased the expression and activity of catalase, but not that of SOD and GPx in vascular adventitial fibroblasts [4]. And, a subpressor dose of Ang II elevated blood pressure and decreased SOD activity, but no significant change to GPx [5]. Based on the various reports presented above, it seems that the data of the activities of catalase and SOD including NADPH oxidase are enough in Ang II-stimulated H9C2 cells, although the glutathione peroxidase activity was not determined in this study. However, we shortly described it in "Discussion" on page 8 line 273-276. Two references (reference numbers, 45 and 46) were added in "Reference list" on page 13 line 507-511.

Figure A. Modulation of reactive oxygen species (ROS) by Lubos et al. Antioxid. Redox Signal. 2015, 15, 1957-1997.

Reference 1: Godin, N.; Liu, F.; Lau, G.J.; Brezniceanu, M.L.; Chenier, I.; Filep, J.G.; Ingelfinger, J.R.; Zhang, S.L.; Chan, J.S. Catalase overexpression prevents hypertension and tubular apoptosis in angiotensinogen transgenic mice. Kidney Int. 2010, 77, 1086-1097.

Reference 2: Yang, W.; Zhang, J.; Wang, H.; Gao, P.; Singh, M.; Shen, K.; Fang, N. Angiotensin II downregulates catalase expression and activity in vascular adventitial fibroblasts through an AT1R/ERK1/2-dependent pathway. Mol. Cell Biochem. 2011, 358, 21-29.

Reference 3: Lijnen, P.J.; van Pelt, J.F.; Fagard, R.H. Downregulation of manganese superoxide dismutase by angiotensin II in cardiac fibroblasts of rats: Association with oxidative stress in myocardium. Am. J. Hypertens. 2010, 23, 1128-1135.

Reference 4: Lubos, E.; Loscalzo, J.; Handy, D.E. Glutathione peroxidase-1 in health and disease: from molecular mechanisms to therapeutic opportunities. Antioxid. Redox Signal. 2011, 15, 1957-1997.

Reference 5: Govender, M.M.; Nadar, A. A subpressor dose of angiotensin II elevates blood pressure in a normotensive rat model by oxidative stress. Physiol. Res. 2013, 64, 153-159.

7.  Page 3 line 130. Authors wrote “We first confirmed that two single components [roseoside (Roseo) and icariside E4 (IE4)] among five components  had  activity (data not shown).”

Did the Authors test the activity of the three other components?

Answer: Yes, we tested all five single components for expression of AT1. The three single components (isolariciresinol 9-O-β-D-xyloside, massonianoside B and nicotiflorin) showed no influence on the expression of AT1, except nicotiflorin, which has weak effect (see Figure A below). Double dose pretreatment (100 μg/mL) of nicotiflorin showed inhibitory responses for AT1 expression less than that in roseoside or icariside E4 (50 μg/mL) (see data for roseoside or icariside E4 in Figure 3 in content). We described it in "Results" on page 3 line 130-131.

Figure B. Effects of single component isolariciresinol, massonianoside B or nicotiflorin on the expression AT1 in the Ang II-stimulated H9C2 cells. H9C2 cells (1 x 106 cells) were stimulated with 300 nM Ang II for 7 h. Telmisartan (Telmis, 10 μM) or three single components (each 50, 70 or 100 μg/mL) was pretreated 1 h before Ang II stimulation. Expression of AT1 was determined in mRNA extracts isolated from H9C2 cells using RT-PCR. NC, negative control; Ang II, angiotensin II stimulation; AT1, angiotensin II receptor 1. #, Numbers below band images show the mean values (n = 4) obtained from four independent experiments performed in triplicates.

Reviewer 2 Report

The authors evaluated the effect of NA and the isolated natural compounds roseoside and icariside E4 on AII-induced expression of AT1 receptor and hypertension-related molecules (TNF-alpha, MPCP-1 and TGF-beta) as well on the activity of NADPH oxidase, catalase and SOD and on the levels of hydrogen peroxide and superoxide anion. They isolated and identified the pure compounds from 1.9 kg of NA. All the experiments were performed in cardiomyocytes H9c2 cell line in culture. As a positive control they used telmisartan, an AT1 blocker. The experiments were well designed performed. The conclusions are supported by the data presented.

-Title and abstract: all the abbreviations must be defined the first time used.

- Abstract: It is not correct to state “hypertension” in the following phrase: “…NA has inhibitory effects on the hypertension in Angiotensin II (ANGII)-stimulated HPC2 cells.” This is due to the fact the experiments were performed in cells culture in vitro and not in whole animals.

-Line 48: It is better: “”…vasoactive peptide of the renin-angiotensin system (RAS).”

-Figure 1 (pages 4 and 5): It should not be separated in two pages. 

-Figure 4A, 4D: Please clarify the units in both cases.

-Figure 4B: Please clarify the complete units, i.e how the H2O2 levels were normalized (mg of protein, amount of cells?)

-Line 248: A space should be inserted in  “…withCO2,…”

-Line 292: A space should be inserted in  “…1h…”

- Lines 301 and 303: please put together: 94 °C, 51 °C and 72 °C” 

-Lines 313 and 314: It is better: “Activity of NADPH oxidase, superoxide dismutase and catalase and levels of hydrogen peroxide and superoxide anion”

-All figures: Please verify that all abbreviations used in the figures are defined in the respective figure legend.

- It is better “O2·¾than “·O2¾

Author Response

Reviewer 2.

Comments and Suggestions for Authors

The authors evaluated the effect of NA and the isolated natural compounds roseoside and icariside E4 on AII-induced expression of AT1 receptor and hypertension-related molecules (TNF-alpha, MPCP-1 and TGF-beta) as well on the activity of NADPH oxidase, catalase and SOD and on the levels of hydrogen peroxide and superoxide anion. They isolated and identified the pure compounds from 1.9 kg of NA. All the experiments were performed in cardiomyocytes H9c2 cell line in culture. As a positive control they used telmisartan, an AT1 blocker. The experiments were well designed performed. The conclusions are supported by the data presented.

Title and abstract: all the abbreviations must be defined the first time used.

Answer: The "AT1" used in "Title" was defined as "angiotensin II receptor 1".

The TNF-ɑ, MCP-1, TGF-b, NADPH, H2O2, or ×O2- in the "Abstract" was defined in full name on page 1 line 33-36.

2. Abstract: It is not correct to state “hypertension” in the following phrase: “…NA has inhibitory effects on the hypertension in Angiotensin II (ANGII)-stimulated HPC2 cells.” This is due to the fact the experiments were performed in cells culture in vitro and not in whole animals.

Answer: We changed it to "hypertension-related molecules" on page 1 line 26.

Line 50: It is better: “”…vasoactive peptide of the renin-angiotensin system (RAS).”

 Answer: We added "the" at the front of renin-angiotensin system on page 2 line 50.

Figure 2 (pages 4 and 5): It should not be separated in two pages. 

 Answer: Figure 2 was shown on the same page of page 5.

Figure 4A, 4D: Please clarify the units in both cases.

 Answer: We clarified the figure units in Figure 1B, E and Figure 4A, D. That is, we originally showed the ratio for negative control as "1". Now, the revised figures were based on the percentage.

Figure 4B: Please clarify the complete units, i.e how the H2O2 levels were normalized (mg of protein, amount of cells?)

 Answer: The unit of H2O2 was clarified as "pmol/1 x 106 cells", and it was indicated in Figure 1C on page 4 and Figure 4B on page 7.

Line 314: A space should be inserted in  “…with CO2,…”

 Answer: We inserted a space between ¢with¢ and "CO2" on page 9 line 314.

Line 322-323: A space should be inserted in  “…1h…”

 Answer: We inserted a space between "1" and "h" on page 9 line 322-323.

9. Lines 333: please put together: 94 °C, 51 °C and 72 °C” 

 Answer: We could not understand the "put together: 94 °C, 51 °C and 72 °C" on page 9 line 333. Please let me know what it means.

Lines 344-345: It is better: “Activity of NADPH oxidase, superoxide dismutase and catalase and levels of hydrogen peroxide and superoxide anion”

 Answer: We indicated it as requested on page 9 line 344 and 345.

All figures: Please verify that all abbreviations used in the figures are defined in the respective figure legend.

 Answer: All abbreviations used in the figures were defined in all the figure legends.

12. It is better “O2·¾”than “·O2¾

 Answer: We showed the correct symbols for superoxide anion as "×O2-", and we did not find the “O2·¾” symbols in our manuscript. Please check again and let me know about it.

Round 2

Reviewer 1 Report

In the revised version of their MS, Authors took into account all points I raised.